# Genome-Wide Characterization of the Phenylalanine Ammonia-Lyase Gene Family and Their Potential Roles in Response to *Aspergillus flavus* L. Infection in Cultivated Peanut (*Arachis hypogaea* L.)

**DOI:** 10.3390/genes15030265

**Published:** 2024-02-21

**Authors:** Pengpei Chai, Mengjie Cui, Qi Zhao, Linjie Chen, Tengda Guo, Jingkun Guo, Chendi Wu, Pei Du, Hua Liu, Jing Xu, Zheng Zheng, Bingyan Huang, Wenzhao Dong, Suoyi Han, Xinyou Zhang

**Affiliations:** 1The Shennong Laboratory/Postgraduate T&R Base of Zhengzhou University, Xinxiang 453500, China; 17862705335@163.com (P.C.); cui2015104035@163.com (M.C.); zq43901997@163.com (Q.Z.); gtd8586@163.com (T.G.); 17631525928@163.com (J.G.); 18251908980@163.com (C.W.); dupei2005@163.com (P.D.); 2Institute of Crop Molecular Breeding, Henan Academy of Agricultural Sciences/Key Laboratory of Oil Crops in Huang-Huai-Hai Plains, Ministry of Agriculture/Henan Provincial Key Laboratory for Oil Crop Improvement, Zhengzhou 450002, China; clj18336079086@163.com (L.C.); liuhua703@126.com (H.L.); xj2000198@126.com (J.X.); zhengzheng@hnagri.org.cn (Z.Z.); huangbingyan@aliyun.com (B.H.); dongwzh@126.com (W.D.)

**Keywords:** phenylalanine ammonia-lyase, peanut, *A. flavus*, phylogenetic analysis, cis-element analysis, expression analysis

## Abstract

Phenylalanine ammonia-lyase (PAL) is an essential enzyme in the phenylpropanoid pathway, in which numerous aromatic intermediate metabolites play significant roles in plant growth, adaptation, and disease resistance. Cultivated peanuts are highly susceptible to *Aspergillus flavus* L. infection. Although *PAL* genes have been characterized in various major crops, no systematic studies have been conducted in cultivated peanuts, especially in response to *A. flavus* infection. In the present study, a systematic genome-wide analysis was conducted to identify *PAL* genes in the *Arachis hypogaea* L. genome. Ten *AhPAL* genes were distributed unevenly on nine *A. hypogaea* chromosomes. Based on phylogenetic analysis, the *AhPAL* proteins were classified into three groups. Structural and conserved motif analysis of *PAL* genes in *A. hypogaea* revealed that all peanut *PAL* genes contained one intron and ten motifs in the conserved domains. Furthermore, synteny analysis indicated that the ten *AhPAL* genes could be categorized into five pairs and that each *AhPAL* gene had a homologous gene in the wild-type peanut. Cis-element analysis revealed that the promoter region of the *AhPAL* gene family was rich in stress- and hormone-related elements. Expression analysis indicated that genes from Group I (*AhPAL1* and *AhPAL2*), which had large number of ABRE, WUN, and ARE elements in the promoter, played a strong role in response to *A. flavus* stress.

## 1. Introduction

The phenylpropanoid pathway, one of the important secondary metabolic ways, widely exists in higher plants and some microorganisms [1]. All substances containing the phenylpropanoid skeleton are direct or indirect products of this pathway. This route produces a wide range of aromatic metabolites, including phytoalexin, lignin, flavonoids, iso-flavonoids, anthocyanins, and phytohormones [2], all of which are crucial for plant growth, development, and their resilience to biotic and abiotic stressors [3,4,5]. Phenylalanine ammonia-lyase (PAL, EC4.3.1.5), part of the ammonia-lyase superfamily, is the initial enzyme of this pathway and serves as a pivotal rate-limiting catalyst. It catalyzes the deamination of L-phenylalanine to *trans*-cinnamic acid, which is responsible for diverting carbon flux from the primary metabolic pathway to the trans-cinnamic acid synthesis pathway, and then further converts into different secondary metabolites in plants [2,6].

Numerous studies have indicated that the count of *PAL* family members is relatively limited. However, this number exhibits significant variation across different plant species. In *Arabidopsis thaliana*, four *PAL* genes have been identified [7], while *Scutellaria baicalensis* and *Coffea canephora* each have three [8,9], and poplar possesses five [10]. Twelve *PAL* members have been identified in rice and watermelon, 13 in cucumber [11,12], 16 in grapevines, and more than 20 in potato and tomato [13,14,15,16]. Functional differentiation results from duplication events in the *PAL* genes of plants. *PAL* families in plants likely originate from gene duplications, such as tandem, segmental, and whole-genome duplications [11]. And *PAL* isoforms play unique roles in plant development. For instance, *Arabidopsis thaliana AtPAL1*, *AtPAL2*, and *AtPAL4* are strongly expressed in inflorescent stems, a tissue rich in lignifying cells, whereas the *AtPAL3* transcript is expressed at an exceptionally low level, implying a role for *AtPAL1*, *AtPAL2*, and *AtPAL4* in tissue-specific lignin synthesis [7]. Phenotypic analysis of single and multiple mutants of *AtPAL* genes revealed that *AtPAL1* and *AtPAL2* might have redundant roles in *Arabidopsis thaliana* flavonoid biosynthesis. However, *AtPAL1* and *AtPAL2* double mutants are more tolerant to drought conditions than wild-type plants [3]. In raspberry (*Rubus idaeus*) plants, *RiPAL1* is associated with early fruit ripening, whereas *RiPAL2* expression correlates more with later stages of flower and fruit development, although the two genes are similar in sequence [17]. In *Salvia miltiorrhiza*, *SmPAL1* and *SmPAL3* are highly expressed in roots and leaves, whereas *SmPAL2* is predominantly expressed in stems and flowers, indicating that *SmPAL1* and *SmPAL3* function redundantly in rosmarinic acid biosynthesis [18]. What is more, it has been reported that *PAL* genes play a role in response to pathogenic infections. For example, of the four *Arabidopsis PAL* genes, *AtPAL1* and *AtPAL2* are co-expressed in different plant organs in response to inductive treatment with phytopathogenic *Pseudomonas*. Additionally, in response to nitrogen depletion, *AtPAL1* and *AtPAL4* transcript levels significantly increase [19]. In beans, *GmPAL1* and *GmPAL3* can be induced by fungal infection [20], and in pepper, *CaPAL1* acts as a positive regulator of salicylic acid-dependent (SA-dependent) defense signaling to combat microbial pathogens [4].

Cultivated peanuts (*Arachis hypogaea* L.) are one of the crops most susceptible to *A. flavus* infection, and aflatoxins with strong toxicity and carcinogenicity are produced rapidly after infection, seriously threatening food safety and human health. Mining of regulatory genes related to *A. flavus* stress was considered as an important way to fundamentally solve the problem. It was reported that the fungal components, *Aspergillus flavus* L., induce a defense response in peanut varieties, resulting in increased PAL activity [21]. And polyhydroxylated flavonoids and intermediate products of the phenylpropanoid pathway were found to inhibit covalent adduct formation between the aflatoxin B1 and DNA, suggesting that *PAL* plays a unique role in resistance to *A. flavus* [22]. However, there are no reports on the potential regulatory role of *PAL* in peanut response to *A. flavus* infection. And the genome-wide bioinformatic analysis of the *PAL* gene family in cultivated peanut is still yet to be carried out. The genome sequence of *A. hypogaea* has recently been completed and released [23,24], providing an important resource for genome-wide analysis of disease-related genes. In the present study, ten *AhPAL* genes were identified and analyzed using bioinformatic approaches. Furthermore, the expression levels of *AhPAL* genes in cultivated peanut kernels were calculated at different time points under *A. flavus* stress using available RNA-seq datasets (accession number: PRJNA825125) and RT-qPCR. The findings of this research offer valuable perspectives on the evolutionary trajectory of *PAL* genes and elucidate their adaptive roles when confronting *A. flavus*-induced stress in peanut kernels.

## 2. Material and Methods

### 2.1. Database Searching and Identification of PAL Genes in Cultivated Peanut Genome

The genome sequences of cultivated peanuts (*A. hypogaea* cv. Tifrunner) were retrieved from the National Center for Biotechnology Information (NCBI) database under the accession number GCA_003086295.2 (available at: https://www.ncbi.nlm.nih.gov/assembly (accessed on 10 January 2023)). We used the established PAL sequences from *Arabidopsis thaliana* and *Cucumis sativus* as references, which were procured from TAIR at https://www.arabidopsis.org/index.jsp (accessed on 12 January 2023)and the CuGenDBv2 http://cucurbitgenomics.org/v2/ (accessed on 13 January 2023), respectively. These sequences served as probes for the Basic Local Alignment Search Tool (BLAST) to identify homologous *PAL* genes within the ‘Tifrunner’ genome. Proteins lacking MIO domains were removed via manual examination [25]. Physicochemical profiling of *PAL* genes was performed using ExPASy [26]. Subcellular localization analysis of all identified PAL proteins was conducted using the Plant-Ploc server (http://www.csbio.sjtu.edu.cn/bioinf/plant/ (accessed on 18 January 2023)).

### 2.2. Phylogenetic Analysis of PAL Proteins

The PAL amino acid sequences were retrieved from the reference genome of the cultivated peanut, *A. hypogaea* cv. Tifrunner, accessible at https://www.peanutbase.org/ (accessed on 20 January 2023). We performed multiple sequence alignments utilizing DNAMAN V9 and ClustalX 2.1 software [27,28]. Phylogenetic trees were then generated via MEGA 7.0 software, employing the neighbor-joining model with 1000 bootstrap replications [29]. The following PAL amino acid sequences in dicotyledonous and monocot plants were used for phylogenetic analysis: *Arabidopsis thaliana* (NP_181241, NP_190894, NP_196043, NP_187645), *Nicotiana tabacum* (BAA22963, ACJ66298, AAA34122, CAA55075), *Cucumis sativus* (Csa1M590300.1, Csa4M008760.1, Csa4M008780.1, Csa6M147460.1, Csa6M405960.1, Csa6M445240.1, Csa6M445750.1, Csa6M445760.1, Csa6M445770.1, Csa6M446290.1), *Vitis vinifera* (XP_002285277, XP_002281799, XP_003633986, CAN77065), *Oryza sativa* (BAD23149, BAD23155, CBC40318, CBC40320), and *Zea mays* (AAL40137, NP_001147922, NP_001147433, NP_001151482) [11,12,15].

### 2.3. Chromosome Localization, Gene Structure, Conserved Motif, and Duplication Event Analysis of AhPALs

Information regarding the chromosome localization and the exon–intron structure was obtained from the peanut reference genome (*A. hypogaea* cv. Tifrunner, https://www.peanutbase.org/ (accessed on 25 January 2023)). The conserved domains of AhPALs proteins were extracted using NCBI-Batch-CDD software. Chromosome localization and conserved motifs were identified and presented by the methods described by Cui et al. [30]. Representation of tandem and segmentally duplicated gene pairs and circos figures of PAL duplication links were conducted using TBtools software [31].

### 2.4. Cis-Acting Element Analysis of AhPALs

Approximately 2000 bp sequences upstream of the transcription start site (TSS) of the *AhPAL* genes were analyzed to identify potential cis-regulatory elements. The analysis utilized the Plant CARE online resource (available at http://bioinformatics.psb.ugent.be/webtools/plantcare/html/, accessed on 25 November 2022) for the identification of these elements [32]. Visualization of the findings was accomplished through TBtools.

### 2.5. In Silico Expression Analysis of AhPAL Genes Using RNA-seq Data

RNA-seq data from *A. flavu*s-infected cultivated peanut samples were retrieved from the NCBI’s Sequence Read Archive (SRA) database (Accession No. PRJNA825125) [33]. We quantified the transcript levels of the *AhPAL* genes across different samples using the Fragments Per Kilobase Million (FPKM). Heat maps generated via TBtools illustrated the differential expression patterns of the *AhPAL* genes.

### 2.6. Plant Materials and Quantitative RT-PCR-Based Expression Assays of AhPALs

In our preceding study [33], we collected samples from kernels of “J-11” (a resistance cultivar) and “Zhonghua 12” (a susceptible cultivar) at various time intervals (0, 1, 3, 5, and 7 days) following inoculation with *A. flavus* spores. Total RNA was isolated from the collected kernels using the RNAprep Pure Plant Plus Kit (Tiangen Biotech, Co., Beijing, China) according to the manufacturer’s protocol. First-strand cDNA was obtained using the PrimeScrit™ RT Kit with gDNA eraser (perfect real-time, Takara Biomedical Technology, Ltd., Beijing, China). RT-qPCR primers were designed by Primer 3.0 online software (https://bioinfo.ut.ee/primer3/ (accessed on 18 February 2023)), and the alcohol dehydrogenase class III (*AhADH3*, Arahy. VYWU26.2) was selected as the internal reference control [33] (Appendix A). Quantitative RT-qPCR and expression levels of *AhPALs* were conducted using the methods presented in the study by Cui et al. [30].

## 3. Results

### 3.1. Genome-Wide Identification of PAL Genes in the Cultivated Peanut Genome

To accurately retrieve the complete PAL proteins in peanuts, all chromosomes and scaffold sequences in the cultivated peanut genome database (*A. hypogaea* cv. Tifrunner, https://www.peanutbase.org/ (accessed on 10 January 2023)) were searched using PAL proteins from *Arabidopsis* (*AtPALs*, NP_181241) and cucumber (*CsPALs*, Csa6G445760.1) as queries. In this study, we identified 10 MIO domain-containing proteins as PAL members in cultivated peanuts, with their lengths ranging from 695 to 729 amino acids. Analysis of molecular properties revealed that the molecular weights of these *PAL* members span between 75.75 and 79.30 KDa, while their isoelectric points vary from 5.93 to 6.68. Significantly, subcellular localization predications predicted that all 10 *PAL* genes in cultivated peanuts are cytoplasmic (refer to Table 1).

The ten *AhPAL* genes were mapped to nine chromosomes of *A. hypogaea,* corresponding to their physical positions as depicted in Figure 1. According to their order on the chromosomes, *PAL* genes were renamed as *AhPAL1*–*AhPAL10* (Table 1). Only chromosome 17 contained two *PAL* genes (*AhPAL7* and *AhPAL8*), whereas all other chromosomes contained one PAL member.

### 3.2. Sequence Characterization and Phylogenetic Analysis of PAL Family Proteins in Cultivated Peanut

Detailed sequence alignment was performed for all deduced AhPAL proteins. As has been reported for PAL proteins in cucumber, melon, and watermelon [11,12,15], all AhPAL proteins contained four conserved domains (Figure 2 and Appendix A): the N-terminus (residues 1–22, numbered according to AhPAL1), MIO (residues 23–259), core (residues 260–525 and 643–714), and inserted shielding (residues 526–642). Importantly, the N-terminus of all AhPAL proteins showed the greatest divergence, consistent with PAL families in other plants [11,12,15,17,34]. Additionally, the conserved enzymatic active site, -^200^Ala-Ser-Gly (numbered according to AhPAL1), in the MIO domain and residues that were essential for PAL enzymatic activity (^349^Tyr and ^492^Gly) in the core domain were found in all AhPAL proteins [11,12,15,35]. Another significant residue is ^547^Thr, located within the shielding domain, identified as a potential phosphorylation site [12]. As depicted in Figure 2B, this site is conserved across only six AhPALs isoforms; in contrast, three out of the remaining four PAL proteins exhibit substitutions with Asn and Ser. This pattern suggests that the AhPAL family may exhibit enzymatic activity, albeit with varying degrees of substrate specificity.

To determine the evolutionary relationship among plant PAL proteins, sequences of the PAL family from dicot (Arabidopsis, tobacco, cucumber, and grape) and monocot plants (rice and maize) were analyzed using a neighbor-joining phylogenetic tree. As shown in Figure 3, the corresponding tree categorizes these PALs into monocot and dicot groups. The ten AhPAL proteins were classified into three groups (AhPAL3-AhPAL7-AhPAL8-AhPAL9, AhPAL4-AhPAL10, and AhPAL1-AhPAL2-AhPAL5-AhPAL6) and members of the dicot group (Figure 3 and Appendix A). Moreover, the plant PALs within one species were more closely related to each other than to the homologs in other plants, indicating the functional conservation of PALs in the same plant.

### 3.3. Exon–Intron and Conserved Motif Analysis of AhPALs

The exon–intron structures and conserved motifs of *AhPALs* in *A. hypogaea* were investigated using the Gene Structure Display Serve (GSDS) online software and MEME tool, respectively. Notably, all *AhPAL* genes had one intron in the middle and ten motifs in the conserved domains (Figure 4), indicating that *AhPAL* genes are conserved in *A. hypogaea*.

### 3.4. Duplication Events in AhPAL Genes and Synteny Analysis

Segmental and tandem duplications lead to duplicated gene pairs and play a major role in the expansion and evolution of gene families in various plant species [36]. Ten segmental duplicated gene pairs involving eight *AhPAL* gene members were identified in the *A. hypogaea* genome (Figure 5 and Appendix A). However, no tandem duplications occurred, indicating that segmental duplication may be the major driving force for the expansion of the *PAL* gene family in the *A. hypogaea* genome. Similar findings have been reported for the AP2/ERF [30], WRKY [37], GRFs [38], MST [39], and bHLH [40] gene families of *A. hypogaea*. It is noteworthy that all duplicated gene pairs were clustered in the same group based on phylogenetic analysis, implying that the gene structure was relatively conserved during evolution.

Furthermore, we investigated the synteny of *PAL* genes in *A. hypogaea*, *A. duranensis*, *A. ipaensis*, *Medicago truncatula,* and soybean genomes to characterize the evolutionary patterns of *AhPAL* genes within the Leguminosae species (Figure 6 and Appendix A). Ten and eleven gene pairs were identified between cultivated peanuts and wild-type *A. duranensis* and *A. ipaensis*, respectively (Appendix A). It appeared that most genomes of *A. duranensis* and *A. ipaensis* might have more than one ortholog in the cultivated peanuts (Appendix A). *A. hypogaea* contained twice the number of PAL members observed for *A. duranensis* and *A. ipaensis*, which is consistent with the fact that *A. hypogaea*, a tetraploid plant, is formed by chromosome doubling after the natural hybridization of two diploid wild species [23,24,41]. Similarly, 11 and 13 synteny gene pairs were detected between *A. hypogaea* and *M. truncatula* and the common bean, respectively (Appendix A). Additionally, nine *AhPAL* genes were orthologous in *M. truncatula*, the dicot plant, whereas seven *AhPAL* genes were found in the common bean, the monocot plant, suggesting that cultivated peanuts have close genetic relationships with other dicotyledonous plants.

### 3.5. Cis-Acting Elements of AhPALs Promoter Region

To further ascertain the potential role of peanut *PAL* genes in various biological processes, the promoter region upstream of 2000 bp of 10 *AhPALs* was investigated [42,43]. Based on the PLANTCARE results, we obtained a total of 43 known cis-acting elements that could be classified into four types: tissue-specific elements (4), stress-related elements (6), hormone-related elements (9), and light-related elements (24) (Appendix A). As shown in Figure 7A, except for *AhPAL2*, *AhPAL3*, *AhPAL7*, *AhPAL8*, and *AhPAL9*, which lack tissue-specific elements, all other *AhPALs* contained the four types of elements in their promoter region. To better explain the regulatory roles of *AhPALs* in plant stress responses, the number and types of hormone- and stress-related elements were counted and are presented in Figure 7B. Among them, the ABRE element responded to abscisic acid; the CGTCA-motif and TGACG-motif were responsive to MeJA; the AuxRR-core and TGA-element responded to auxin; the GARE-motif and TATC-box were responsive to gibberellin; and the TCA-element responded to salicylic acid. The cis-elements related to stress include drought inducibility (MBS), wound response (WUN), low-temperature (LTR), anaerobic induction (ARE), and TC-rich elements. Among these cis-elements, ABRE and MeJA were the two most frequently identified elements, followed by MBS, ARE, and WUN, indicating that most *AhPAL* genes participate in the regulation of plant stress processes by responding to hormones.

### 3.6. Expression Patterns of AhPAL Genes in Response to A. flavus Stress in Peanut Seeds

To better investigate the possible regulatory roles of *AhPALs* in peanuts’ response to *A. flavus* infection, the expression patterns of *AhPAL* genes were analyzed at different time points under *A. flavus* stress in cultivated peanut kernels using the available RNA-seq datasets (accession number: PRJNA825125). Among them, the expression levels of six *AhPAL* genes (four genes from group I and two from group II) were higher in the *S* genotype than in the *R* genotype, implying a main role in the response to *A. flavus* infection (Figure 8A). As shown in Appendix A, profuse mycelial growth and sporulation was observed in the *S* genotype on the 7th day after inoculation, whereas very little mycelial growth occurred on the *R* seed coat. According to the methods described by Cui et al. [33], the infection index was calculated on the 7th day after inoculation and was nearly 5-times higher in *S* (92.22) than that in *R* (17.00). Furthermore, two genes from each group, *AhPAL1* and *AhPAL2* from Group I, *AhPAL3* and *AhPAL8* from Group II, and *AhPAL4* and *AhPAL10* from Group III, were selected to detect whether their expression levels would be induced under *A. flavus* stress by RT-qPCR. As shown in Figure 8, the expression levels of all *AhPALs* genes were higher in *S* than in *R*, suggesting that all *AhPALs* genes were significantly induced by *A. flavus* infection, which may be due to more serious damage in *S*. Moreover, *AhPAL1* and *AhPAL2* from Group I and *AhPAL4* from Group IIII exhibited continuous increasing trends in overall time points in both *R* and *S* kernels. And the expression level of *AhPAL1* and *AhPAL2* in *S* were nearly 10-times higher than that in *R*, while *AhPAL4* only 5-times, implying that genes from Group I functioned strongly upon *A. flavus* stress.

## 4. Discussion

PAL, a member of the ammonia-lyase superfamily, is widely found in higher plants. The characterization of PAL genes in certain plants is crucial, as these genes have been shown to be closely related to biotic and abiotic resistance in other studies [3,4]. The released genome sequence of *A. hypogaea* facilitated identification and isolation of the *PAL* gene in cultivated peanuts. In the present study, ten *PAL* genes in cultivated peanuts were identified using BLAST. The analysis of PAL proteins translated from these genes showed basic properties, such as isoelectric point and amino acid length varying within a small range, implying that *PAL* genes in the cultivated peanut were conserved and had similar important functions. The subcellular localization predicted that PAL proteins are localized in the cytoplasm, which is consistent with reports in other plants [44,45]. Phylogenetic analysis showed that *PAL* genes from plants were categorized into monocot and dicot groups. Ten AhPAL proteins were classified into three groups, similar to that found in walnut [46], whereas most woody plants have only two groups [12,45,47,48], suggesting that the *PAL* gene was relatively conserved in dicot plant evolution. Moreover, gene structure analysis showed that all *AhPAL* genes contained one intron in the middle and ten motifs in the conserved domains (Figure 4), indicating that *AhPAL* genes are conserved in *A. hypogaea*.

Gene duplication events include whole-genome fragment duplication, small-scale fragment replication, partial tandem replication, or a combination of these [49]. The ten *PAL* genes can be divided into five pairs, of which four pairs correspond one-to-one in their two chromosome groups (chr02-chr12, chr06-chr16, chr08-chr18, and chr09-chr19). Excluding one gene pair (chr8-chr18), the other three gene pairs were also similar in position and may have been produced by the duplication of whole-genome segments. *AhPAL3* and *AhPAL9* are located in the middle and top of their chromosomes, respectively, which could be due to the replication of whole-genome fragments, followed by chromosome recombination. Cultivated peanuts included two sub-genomes: A and B. In the peanut reference genome, chr01 to chr10 correspond to the A sub-genome and chr11 to chr20 correspond to the B sub-genome [41]. These four gene pairs were equivalent to a one-to-one correspondence in the A and B sub-genomes. Studies have shown that cultivated peanuts may have resulted from the fusion of two diploid wild-type peanuts [50]. Comparing the cultivated peanut *PAL* with that of the wild-type peanut, it was observed that each *PAL* gene of the cultivated peanut corresponded to the *PAL* from either of the wild-type peanut *PAL* genes (Figure 6 and Appendix A).

The phenylpropanoid pathway produces a vast number of aromatic metabolites, including flavonoids, isoflavonoids, anthocyanins, plant hormones, phytoalexins, and lignins, and PAL activity is related to the productivity of these products. Transcriptome research showed that *PAL* gene expression in resistant seeds differed from that in susceptible seeds during *A. flavus* infection [45,51,52]. It was also found that phenylpropanoid biosynthesis pathways are triggered by transcription factors, such as *WRKY*, *bHLH*, and *MYB*, to evoke defense response mechanisms against *A. flavus* [53]. The expression profile of *AhPAL* genes in peanuts infected with *A. flavus* showed that all *AhPALs* genes exhibited a higher level in *S* than that in *R*, suggesting that all *PALs* genes were significantly induced by *A. flavus* infection, which may due to more serious damage and more mycelia and spores in *S*. Moreover, the spores on the surface of the kernel increased during *A. flavus* infection process, combing the fact that expression levels of *AhPALs* exhibited continuous increasing trends in overall time points, implying that expression of *PAL* genes is closely related to mycelial growth. Meanwhile, the expression levels of *AhPAL1* and *AhPAL2* in *S* were nearly 10-times higher than that in *R*, suggesting its critical role in response to *A. flavus* infection. The promoter sequence was regarded as a major factor determining whether and when the transcription of a gene will be initiated and possessing cis-acting elements, which might imply the potential functions of genes [30,54]. Here, genes from Group I showed the largest number of ABRE, WUN, and ARE in the promoter region, in combination with the phylogenetic tree, illustrating that the phylogenetically similar genes shared identical cis-elements, and increased number of ABRE, WUN, and ARE elements correspond to an increased responsiveness to *A. flavus* infection.

## 5. Conclusions

This study conducted a comprehensive identification and characterization of the peanut *PAL* gene family at the genome level. We identified ten *PAL* genes in cultivated peanuts, which were unevenly distributed across nine chromosomes. Phylogenetic analysis revealed that dicot *PAL* genes segregate into three distinct clades. The pattern of chromosome distribution and synteny analysis suggest that segmental duplications played a significant role in the proliferation of the peanut *PAL* gene repertoire. Notably, genes within Group I (*AhPAL1* and *AhPAL2*), which possess a high density of ABRE, WUN, and ARE elements within their promoter regions, exhibited pronounced activity in response to *A. flavus* stress.

## Figures and Tables

**Figure 1 genes-15-00265-f001:**
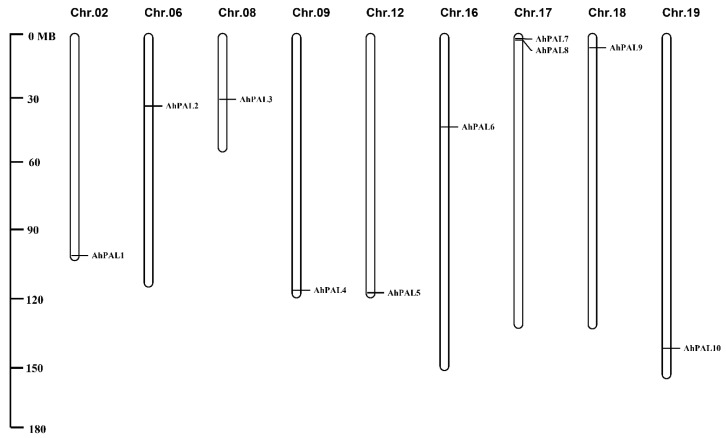
Chromosome distributions of *AhPAL* genes in cultivated peanut with the chromosome number indicated at the top of each representation.

**Figure 2 genes-15-00265-f002:**
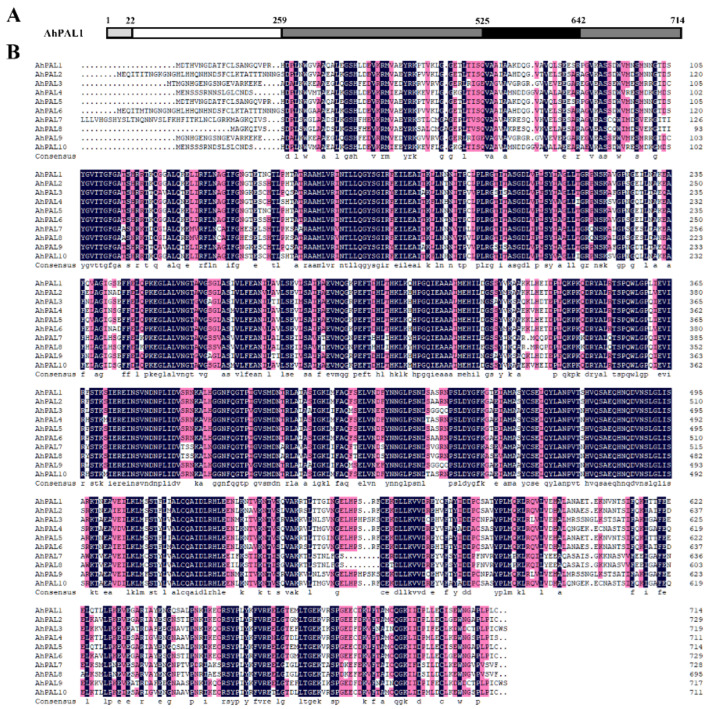
Sequence analysis of AhPAL proteins. A schematic diagram of AhPAL amino acid sequence is presented in panel (**A**). AhPAL1 is depicted as a representative of the PAL protein families. The diagram indicates the four functional domains: the N-terminal domain (light gray), MIO domain (white), core domain (dark gray), and inserting shielding domain (black). Panel (**B**) shows the sequence alignment of AhPAL proteins, which was performed using DNAMAN V6. Gaps in the alignment are represented by dashes.

**Figure 3 genes-15-00265-f003:**
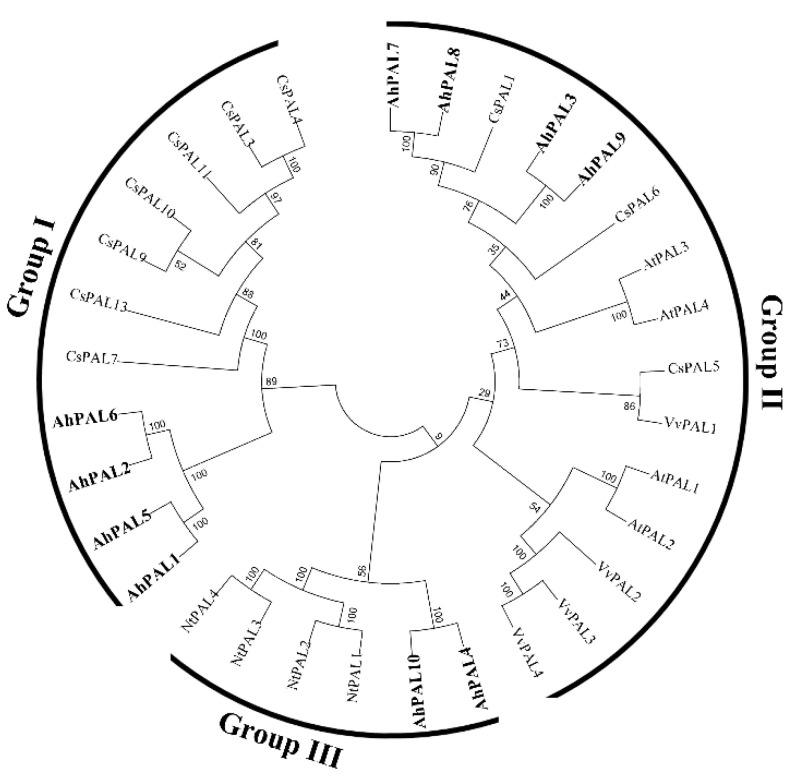
Phylogenetic tree of PAL proteins in plants. The conserved PAL proteins from Arabidopsis (AtPAL), tobacco (NtPAL), cucumber (CsPAL), grape (VvPAL), and cultivated peanut (AhPAL) were aligned using Clustal X. The phylogenic tree was constructed using the neighbor-joining model (1000 replicates) with the MEGA 7.0 program. The AhPAL proteins are highlighted in bold.

**Figure 4 genes-15-00265-f004:**
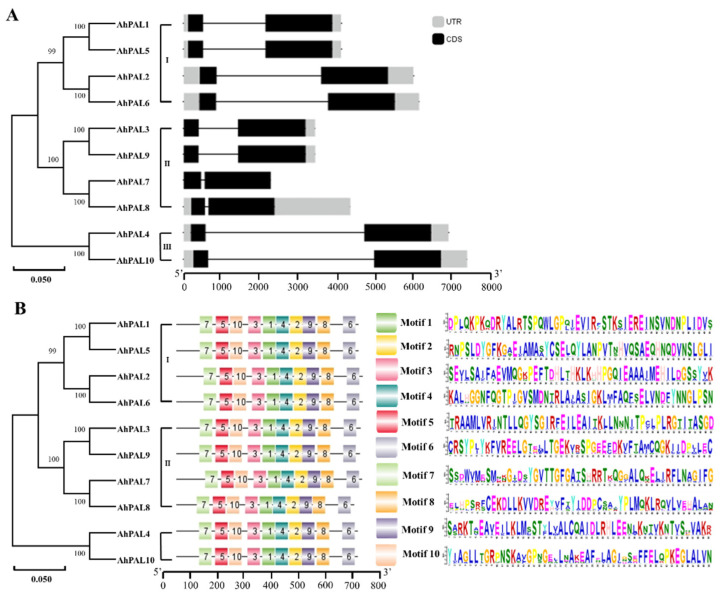
Exon–intron structure and conserved motif analysis of *AhPAL* genes. (**A**) The exon–intron structure of *AhPAL* genes. (**B**) The distribution of conserved motifs in AhPAL proteins.

**Figure 5 genes-15-00265-f005:**
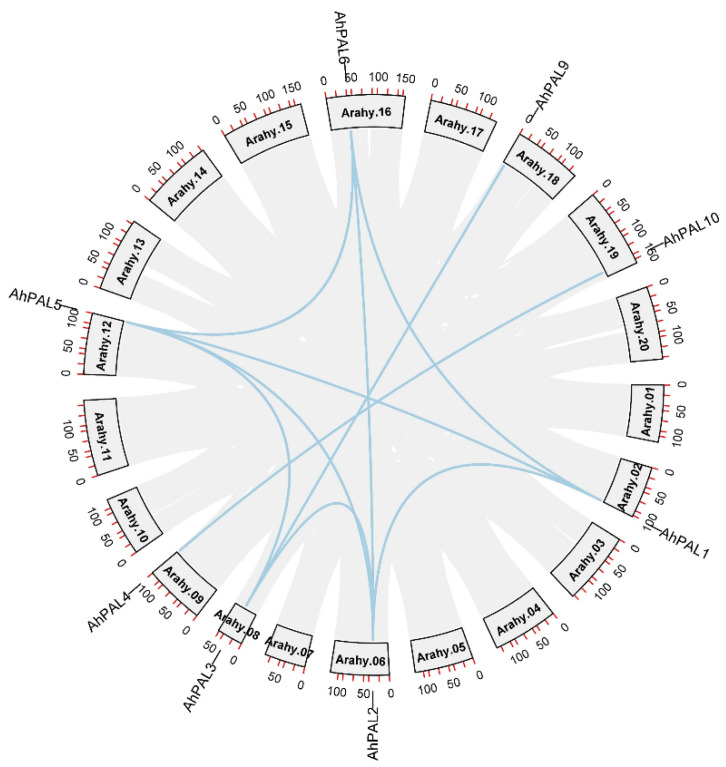
Circos figures for chromosome locations with *AhPAL* segmental duplication links. The blue lines indicate segmental duplicated gene pairs.

**Figure 6 genes-15-00265-f006:**
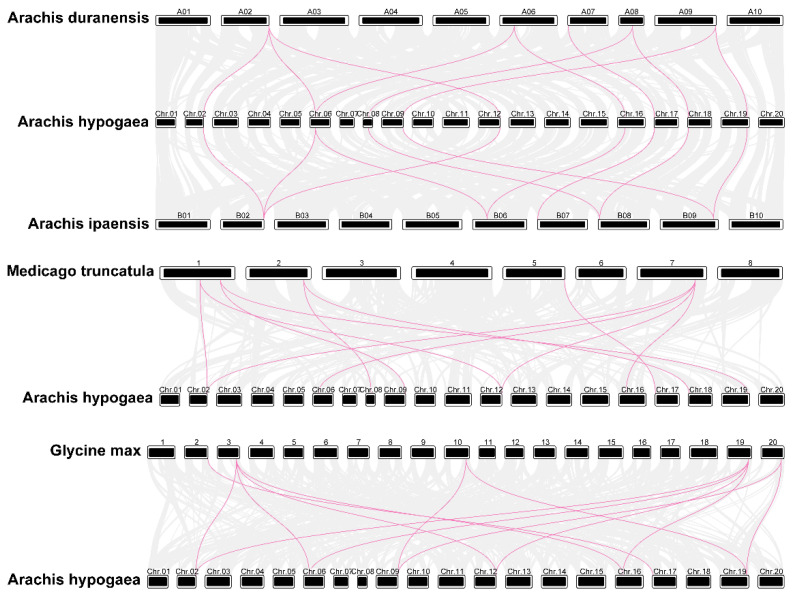
Synteny of *AhPAL* genes in different genomes of *A. duranensis*, *A. ipaensis*, *Glycine max* L., and *A. hypogaea* cv. Tifrunner.

**Figure 7 genes-15-00265-f007:**
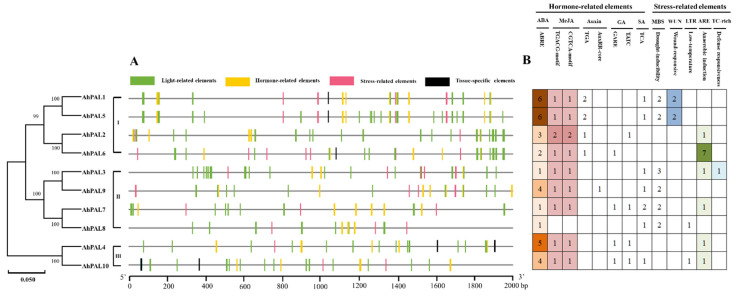
Predicted cis-acting elements of *AhPAL* gene promoters in *A. hypogaea*. (**A**) Promoter regions 2000 bp upstream of ten *AhPALs* were analyzed using PlantCARE. The different colored rectangles represent the four types of cis-elements. (**B**) The number of hormone- and stress-related elements of *AhPALs* in *A. hypogaea*. Different colors represent numbers of various cis-acting elements.

**Figure 8 genes-15-00265-f008:**
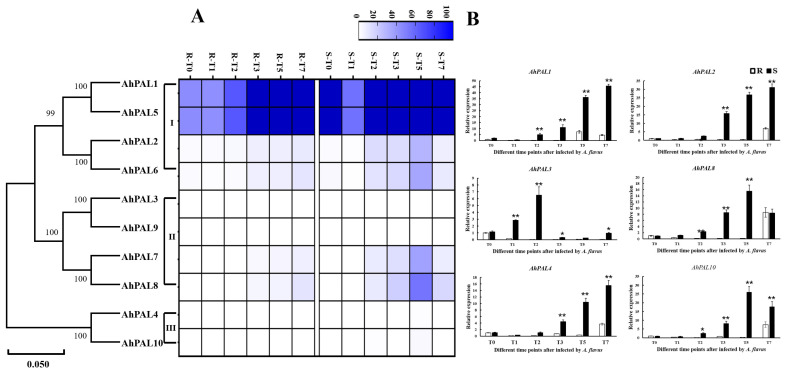
Expression patterns of *AhPAL* genes in *A. hypogaea* cv. Tifrunner. (**A**) The expression patterns of *AhPAL* genes in peanut seeds under *A. flavus* stress. (**B**) Relative expression level of *AhPAL* genes at different time points after infection by *A. flavus*. Error bars were obtained from three biological replicates. Values are means ± standard errors (SEs) of three independent biological replicates (n = 3). Asterisks indicate a significant difference between R and S at each time point as determined by Student’s *t*-test (* *p* < 0.05; ** *p* < 0.01).

**Table 1 genes-15-00265-t001:** Phenylalanine ammonia-lyase (PAL) gene family information in cultivated peanut (*A. hypogaea*).

Gene Name	Gene ID	Subcellular Location	Amino Acid (AA)	Molecular Weight (kDa)	Theoretical pI	Chromosome	Gene Position	
*AhPAL1*	arahy. FI9C9D.1	Cytoplasm	714	77.94	5.96	Arahy.02	102,530,759	102,534,874
*AhPAL2*	arahy. V1SQAY.1	Cytoplasm	729	79.3	6.21	Arahy.06	32,131,529	32,137,537
*AhPAL3*	arahy. 9CK4GV.1	Cytoplasm	720	78.94	6.19	Arahy.08	29,020,167	29,023,604
*AhPAL4*	arahy. 0B4MFB.1	Cytoplasm	711	77.56	6.21	Arahy.09	118,976,212	118,983,138
*AhPAL5*	arahy. 5H4H17.1	Cytoplasm	714	77.94	5.96	Arahy.12	120,128,684	120,132,799
*AhPAL6*	arahy. PUMP6L.1	Cytoplasm	729	79.29	6.15	Arahy.16	42,030,770	42,036,927
*AhPAL7*	arahy. EEZ4Y8.1	Cytoplasm	728	79.7	6.68	Arahy.17	389,497	391,773
*AhPAL8*	arahy. G9Z3VW.1	Cytoplasm	695	75.75	6.04	Arahy.17	1,175,458	1,179,817
*AhPAL9*	arahy. U2YH27.1	Cytoplasm	718	78.55	6.11	Arahy.18	4,683,011	4,686,452
*AhPAL10*	arahy. NB2RRK.1	Cytoplasm	711	77.79	5.93	Arahy.19	146,304,646	146,312,059

## Data Availability

Data are contained within the article or Appendix A.

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
