# Peer review of "Genome-Wide Characterization of the Phenylalanine Ammonia-Lyase Gene Family and Their Potential Roles in Response to *Aspergillus flavus* L. Infection in Cultivated Peanut (*Arachis hypogaea* L.)"

_genes, 2024, doi:10.3390/genes15030265_

Round 1

Reviewer 1 Report

Comments and Suggestions for Authors

General:  I would encourage the authors to state what the objective of the work is and what hypothesis is being tested.  Line 82-87 is a very well written description of what they did, but I don’t see a hypothesis or objective.  

I would encourage the authors to add a couple lines to the intro regarding peanut’s genome – the A&B genomes and possibly any PAL sequences found in the A and B donor species.  You note this indirectly at line 213, but I’d suggest that you make it clear that these species are the putative origins of the A&B genomes of domesticated peanut. 

I may not have understood part of your work correctly, but I’m not sure how your conclusion regrading ‘genome-wide segmental duplication events’ matches with the known allotetraploid origin of peanut.  

Similarly, in the discussion, after you have localized the 10 AhPALs on chromosomes can you infer which parent species they came from?

The only linkage in the present manuscript to A. flavus is the RNAseq work of Cui et al.  That work should be better described here.

Specific points

Line 14:  Maybe it’s just a personal preference, but I don’t think there are any “key” enzymes in a pathway. ALL of the enzymes in a pathway are essential, so none of them can really be any more important than any of the others.

Line 33:  Strange grammar.  

Line 45:  Write out “Four” to start a sentence.

Line 47:  Write out “Twelve” to start a sentence.

Line 51-53:  Clarify again here that the AtPAL genes are Arabidopsis.

Line 65:  Why do you write “PAL1” here instead of “AtPAL1

Line 68:  Again, you’re not consistent with how you’re naming these genes.  Sometimes you use the species abbreviation and sometimes you don’t.

Line 70-71:  This is pretty central to your paper.  You should explain what “fungal components” are involved here.

Line 96:  Explain the significance of MIO sequences.

Line 101-114:  These plants seem quite random.  Why not include PAL amino acid sequences from something a little more closely related like Glycine max or Medicago sativa?

Line 145:  I would suggest the word “predicted” in place of “suggested”.  To be clear, this cytoplasmic localization is what is predicted and there are no actual observations to confirm this.  

Line 235 & 55:  I’m not sure from the present manuscript what you mean by “A. flavus stress” I infer, from the methods that you are building on the RNAseq work of Cui et al (ref #33), but more information needs to be given here about this.

Line 319:  Suggest wording “…implying that increased number of ABRE, WUN, and ARE elements correspond to an increased responsiveness to A. flavus infection”

Comments on the Quality of English Language

Author Response

Thank you for taking the time to review this manuscript. Please find our detailed responses below and the corresponding revisions in the re-submitted files. The modified portions have been highlighted in red.

Reviewer 2 Report

Comments and Suggestions for Authors

Manuscript ID: genes-2826684

Title: Genome-wide Characterization of the Phenylalanine Ammonia-lyase Gene Family and Their Potential Roles in Response to Aspergillus flavus L. Infection in Cultivated Peanut (Arachis hypogaea L.)

In the submitted manuscript, the authors examined Arachis hypogaea L. genome associated with activity of Phenylalanine ammonia-lyase (PAL) using bioinformatics methods. The authors proposed a hypothesis suggesting that the mentioned enzyme plays a role in the defense mechanism in response of A. hypogaea L. to the stress induced by the development of Aspergillus flavus. Consequently, ten AhPAL genes of cultivated peanut kernels were identified and studied. Additionally, the expression level of AhPAL genes was investigated in cultivated peanut kernels at different time intervals under A. flavus stress. In my opinion, this manuscript is a valuable extension of existing knowledge because the results of this study offer valuable insight into the evolutionary aspects of the PAL genes and their role in responding to A. flavus stress in peanut kernels. Overall, the article is well written and organized. Below are some comments and suggestions that may further improve the final manuscript:

1)       A valuable addition to the study could be a more precise correlation between the expression of AhPAL and the amount of fungal biomass, which can be measured using traditional methods such as CFU (Colony Forming Units) or ergosterol - a specific component of fungal cell membranes, being often used as a biomarker for fungal biomass.

2)       The authors also did not consider the possibility of the mycotoxin accumulation. As a result, it is unclear whether gene expression was related to the development of fungal biomass, or the accumulation of mycotoxins. If the authors were able to extend the above-mentioned topics, it would valuable extension of the work.

3)       Lines 69-70: The abbreviation should be entered after using the full name, not the other way around. Instead of "SA dependent (salicylic acid dependent)" it should be "salicylic acid dependent (SA dependent)". The same applies to line 134: FPKM (Fragments Per Kilobase Million) => Fragments Per Kilobase Million (FPKM)

4)       Figure 4c): Legend in Fig. 4c) gives the impression that it is a description of individual lines showing the distribution of conserved motifs in AhPAL proteins. It should be moved below the drawing and placed horizontally, or separated from the presented drawing elements.

Author Response

Thank you for taking the time to review this manuscript. Please find below our detailed responses below and the corresponding revisions highlighted in the resubmitted files.

Reviewer 3 Report

Comments and Suggestions for Authors

In the study, the authors utilized a variety of bioinformatic and in silico tools to establish a framework for the identification and characterization of PAL genes (clusters) whose expression could be linked to aflatoxin contamination in cultivated peanut, a persistent food safety issue. The data generated in this study would provide a valuable starting point for investigations aimed at delineating the aflatoxin contamination if there were in vitro or in vivo data to support the data mining findings reported. The introduction section is very general with limited connection with A flavus infection in cultivated peanut, lacking specific justifications why the study would be necessary.

The discussion section is descriptive and should be thoroughly revised by including laboratory level of data for confirmation. The key issue here is the presence of A flavus on peanut is not uncommon but what could trigger A flavus to produce aflatoxins and what kind of role would PAL genes play in this specific pathway? Otherwise the study is just a meaningless showcase of some bioinformatics tools.

Comments on the Quality of English Language

Minor editing needed.

Author Response

Thank you very much for taking the time to review this manuscript. Please find the detailed responses below and the corresponding revisions highlighted in the re-submitted files.

Round 2

Reviewer 3 Report

Comments and Suggestions for Authors

The authors have addressed reviewers' comments and revised the original version accordingly. Therefore, I would like to recommend the current version for publication.